

**Inland lake temperature initialization via cycling with atmospheric data**
**assimilation**
Stanley G. Benjamin[1], Tatiana G. Smirnova[2,1], Eric P. James[2,1], Eric J. Anderson[3],
Ayumi Fujisaki-Manome[4,5], John G.W. Kelley[6], Greg E. Mann[7], Andrew D. Gronewold[5]
Philip Chu[8], Sean G.T. Kelley[9]
[1]NOAA Global Systems Laboratory, Boulder, CO 80305 USA
[2]Cooperative Institute for Research in Environmental Science (CIRES), University of
Colorado, Boulder, CO 80303 USA
[3]Civil and Engineering Department, Colorado School of Mines, Golden, CO USA
[4]Cooperative Institute for Great Lakes Research (CIGLR), University of Michigan, Ann
Arbor, MI USA
[5]University of Michigan, Ann Arbor, MI USA
[6]NOAA National Ocean Service, Coast Survey Development Laboratory, Durham,
NH  03824 USA
[7]NOAA National Weather Service, White Lake, MI, USA
[8]NOAA Great Lakes Environmental Research Laboratory, Ann Arbor, MI, USA
[9]University of Massachusetts, Department of Mathematics and Statistics,
Amherst, MA, USA
*Correspondence to:*  Stan Benjamin (stan.benjamin@noaa.gov)

**Abstract.**  Application of lake models coupled within earth-system prediction models,
especially for short-term predictions from days to weeks, requires accurate initialization
of lake temperatures.   Here, we describe a lake initialization method by cycling within
an hourly updated weather prediction model to constrain lake temperature evolution.
We compare these simulated lake temperature values with other estimates from
satellite and in situ and interpolated-SST data sets for a multi-month period in 2021.
The lake cycling initialization, now applied to two operational US NOAA weather
models, was found to decrease errors in lake temperature from as much as 5-10K
(using interpolated-SST data) to about 1-2 K (comparing with available in situ and
satellite observations.
**Short summary**
Application of 1-d lake models coupled within earth-system prediction models will
improve accuracy but requires accurate initialization of lake temperatures.   Here, we





describe a lake initialization method by cycling within a weather prediction model to
constrain lake temperature evolution. We compare these lake temperature values with
other estimates and found much reduced errors (down to 1-2 K).   The lake cycling
initialization is now applied to two operational US NOAA weather models.
**1  Introduction**
Inclusion of lake representation into numerical weather prediction (NWP) models has
become increasingly necessary to further improve representation of atmosphere-
surface fluxes of heat and moisture as model grid resolution becomes finer.
Representation of lake physics to provide time-varying lake surface properties (e.g.,
Subin et al, 2012) is essential to improve fluxes of heat, moisture and momentum
between the surface and atmosphere (Hostetler et al, 1993, Thiery et al, 2014).   Lake
representation is part of the overall surface treatment including land-surface models
(LSMs) necessary to accurately model the evolution of the planetary boundary layer.
Lakes are estimated to cover 3.7% of the global non-glaciated land area (Verpoorter et
al, 2014), and they significantly moderate sensible heat and moisture fluxes from this
'land' (i.e., non-ocean) area. Water impoundments (reservoirs) that used to account for
about 6% of these 'lake' areas (Downing et al, 2006) have recently increased to 9%
(Vanderkelen et al, 2021). Initial conditions for both land and lake surface are an
important consideration due to far larger thermal inertia for soil or water than for air.
Consequently, incorrect soil or lake initial conditions can result in erroneous heat and
moisture fluxes that may persist for days and even weeks (e.g., Dirmeyer et al, 2018).
This potential source of error in fluxes is more pronounced for lake areas with far larger
thermal inertia and heat storage than even saturated soils.
In the US, operational NWP models have used coarse-resolution daily SST analyses to
specify the surface water temperatures for the ocean and the Laurentian Great Lakes
for the entire forecast period.  However, given the resolution, the temperatures for bays,
sounds, and smaller non-Great Lakes have been obtained by the interpolation of values
from the ocean and the Great Lakes.   An alternative is to incorporate one-dimensional
(1-d) lake models within NWP models.
Lake representation (via one-dimensional (1-d) models, as in LSMs) within NWP
models is beneficial by providing a first-order accurate lagged effect of the seasonal
variation in temperature, with lake water remaining colder than nearby land in spring
and warmer in autumn. The outcomes are desirable, as described by Balsamo et al,
(2012), for instance by accurately representing increased evaporative fluxes in the fall.
Thus, use of a 1-d lake model improves over land representation by capturing this
slower seasonal response.
However, lake temperature initialization is still a problem. Use of spatial interpolation to
smaller lakes from larger (and deeper) lakes, or from the ocean, for lake initialization
(e.g., Mallard et al, 2015) can exaggerate this seasonal slower response. Shallow lakes





warm more slowly in spring than surrounding land, but more quickly than nearby deeper
lakes. Even in summer, it will take at least 1-2 weeks for 1-d models to adjust from
values interpolated from deeper-lake temperatures to become more realistic for shallow
lakes. Therefore, lake temperature initialization becomes the most important factor to
accurately simulate sensible and latent heat fluxes from lakes for short to medium-range
NWP, more so than the use of the lake model itself. One option to solve the lake
initialization problem is to use a model-based climatology for seasonal variation of lake
temperatures (Balsamo et al (2012) and Balsamo (2013)) using a 1-d lake model forced
by reanalysis data. This technique avoids a new spin-up with each new run, but cannot
capture unique weather regime variations in a given region and time. Another option,
described here, is lake cycling, a cost-free option if the atmospheric conditions are
relatively accurate.

Data assimilation for land-surface fields (e.g., soil temperature, soil moisture, snow
cover, snow water equivalent, snow temperature)has been very beneficial for improved
short-range weather prediction accuracy (e.g., Balsamo and Mahfouf, 2020, Muñoz-
Sabater et al, 2019, Benjamin et al, 2022, others), but lake temperature has not been a
part of this surface data assimilation. In December 2020, the NOAA 13-km Rapid
Refresh (RAP) and 3-km High-Resolution Rapid Refresh (HRRR) implemented an
interactive small-lake multi-layer 1-d lake model, the first NOAA weather models to do
so.  The lake coverages for the HRRR and RAP models are shown in Fig. 1.  The
weather models are coupled with the 10-layer Community Land Model (CLM) version
4.5 lake model, (Subin et al, 2012, Mallard et al, 2015), an option within the community
Weather Research and Forecast model (WRF, Skamarock et al, 2019). The CLM lake
model is a 1-d thermal diffusion model allowing 2-way coupling with the atmosphere.
ECMWF had taken a similar approach earlier to improve their overall surface modeling
treatment by implementing the 2-layer FLake (Freshwater Lake Model) model (Mironov
et al, 2010, Balsamo et al, 2012, Boussetta et al, 2021) into their Integrated Forecast
System (IFS) in 2015. To initialize small-lake temperatures in the RAP and HRRR, all
lake variables have been evolving since summer 2018 depending on the cycled
atmospheric conditions and the lake model physics as discussed in section 4.  The 1-d
lake model cannot represent 3-d hydrodynamical processes in larger bodies of water.
Thus, a second major improvement in 2020 with lake representation in the NOAA 3-km
HRRR model occurred with the implementation of lagged data coupling with the 3-d
hydrodynamic-ice model for the Laurentian Great Lakes as described by Fujisaki-
Manome et al (2020).



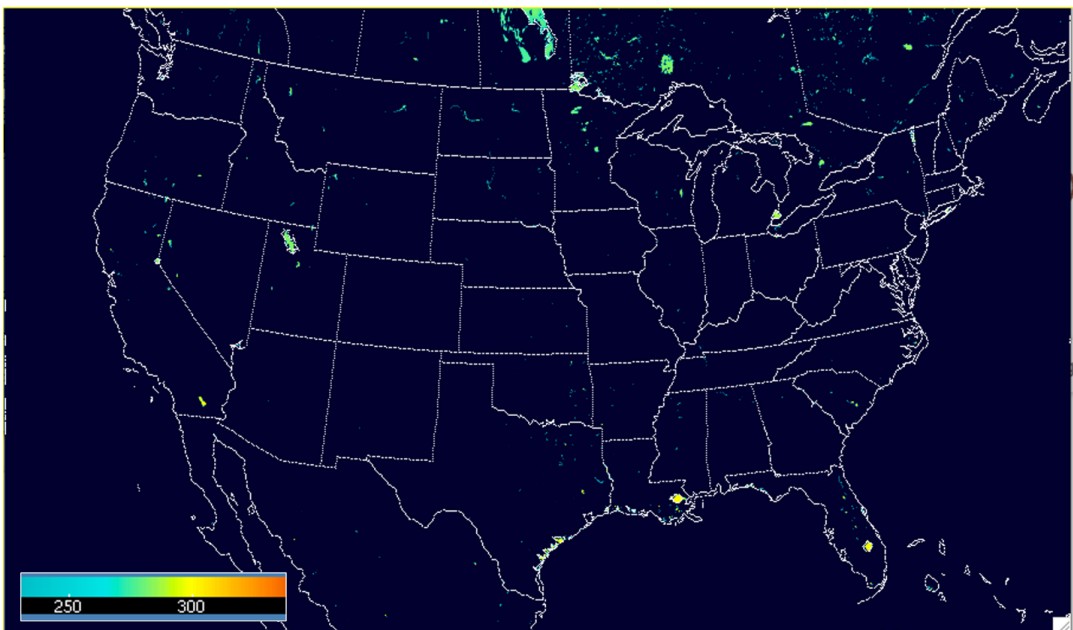

*Fig. 1a.   Small-lake (green to yellow) areas for the a) 3-km HRRR domain using the*
*MODIS 0.15" resolution data for land/water and lake information.  Color corresponds to*
*top-level lake temperature (K) at 01z 15 Oct 2019.    Only small-lake areas treated in*
*HRRR by the CLM lake model are shown.  Out of the 1,900,000 grid points in this*
*HRRR CONUS domain, 12,305 of them (~0.6%) are for small lakes (excluding the 5*
*Laurentian Great Lakes treated by separate coupling as described in text).*



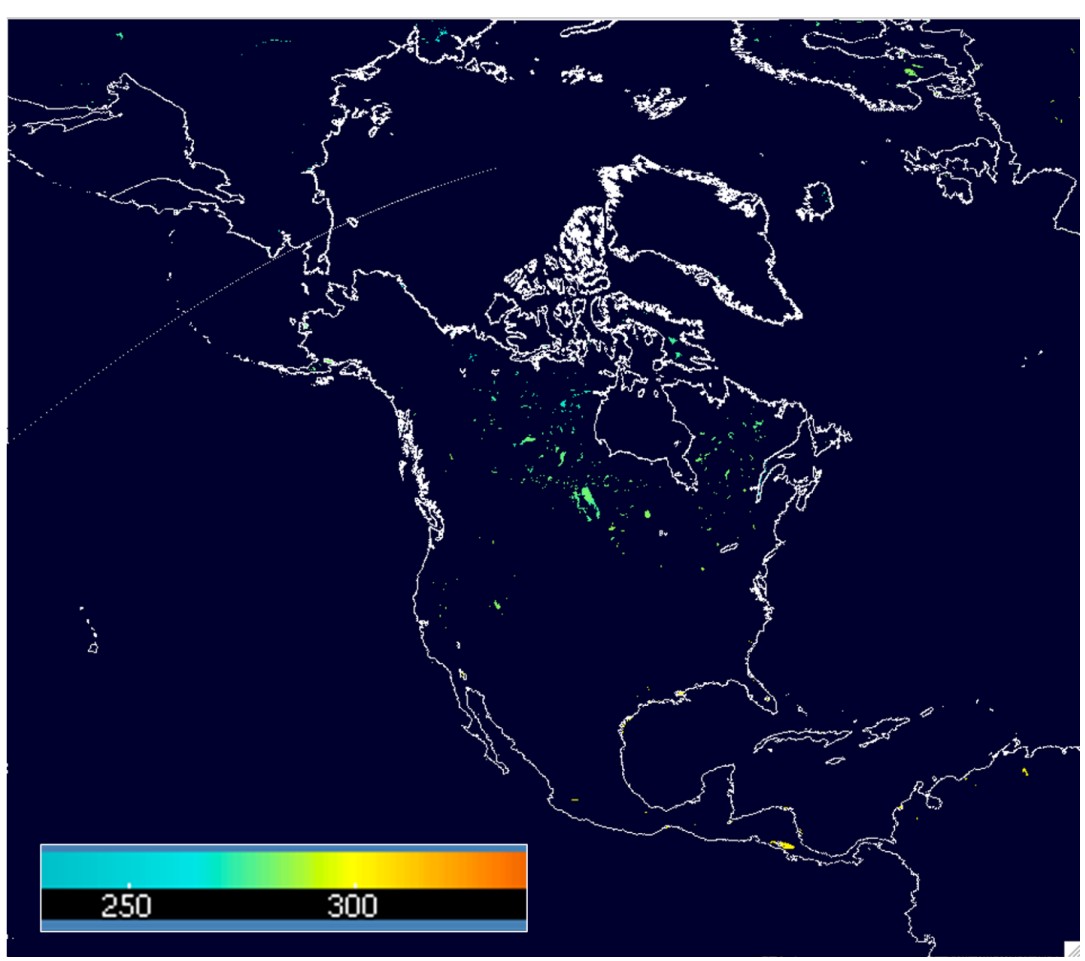

*Fig. 1b.  Same as Fig. 1a but for the 13-km RAP domain*
Here, we describe the design and results of a unique approach to inland small lake
initialization by cycling with hourly updating of atmospheric conditions (clouds/radiation,
near-surface temperature/moisture/winds). This lake initialization via cycling is an
important component of earth-system coupled modeling for effective NWP, with goals to
improve prediction of 2-m (air) temperature and moisture, cloud, boundary-layer
conditions, and precipitation for situational awareness enabling short-range decision
making (e.g., aviation, severe weather, hydrology, energy).
**2    Problem**
For the NOAA hourly updated mesoscale models, used frequently for short-range
weather prediction, poor 2-m air temperature and/or dewpoint forecasts have been
reported on many occasions by the US National Weather Service (NWS) in the vicinity





of inland lakes (Fig. 2). These hourly updated models included the Rapid Update Cycle
(RUC, Benjamin et al 2004) with horizontal grid spacing decreasing from 40-km to 20-
km to 13-km (Benjamin et al 2010), succeeded by the 13-km Rapid Refresh (RAP) and
3-km High-Resolution Rapid Refresh (HRRR, Benjamin et al, 2016, Dowell et al, 2022,
James et al, 2022).  Many of these reported systematic deficiencies from the US NWS
were for the 2.5-km NOAA Real-Time Mesoscale Analysis (RTMA, Pondeca et al.
2011), using 1-h forecasts from the 3-km HRRR as a background. The most common
report was too-low 2-m air temperatures near inland lakes in late spring and summer.
At times, spurious prediction of fog formation was also noted on or near small lakes due
to erroneous lake temperatures and resultant fluxes.

*Fig. 2.  Lakes (here, in black) circled for those with related problem reports from US*
*National Weather Service Forecast Offices on nearby deficient 2-m air temperature or*
*dewpoint forecasts in NOAA hourly updated models during 2004-2019.*
Further investigation revealed the water temperatures for small lakes used in NOAA
weather models were assigned via horizontal interpolation from larger, deeper bodies of
water (with available AVHRR data) in the design for the NOAA real-time gridded SST
analysis (RTG_SST_HR, Gemmill et al, 2007).  An example of the analysis is shown in
Fig. 3. Temperature for the larger, deeper water areas has a lesser and more lagged
seasonal variation than the smaller, shallower lake areas due to their large heat storage
capacity. So use of the NOAA SST fields for lake temperatures resulted in generally
too-low values through spring and summer, and even into autumn. In situations with



atmospheric cold outbreaks in the autumn, shallow lake temperatures quickly decrease
(as reflected with lake cycling) and SST-based estimated lake temperatures were too
high. This behavior was consistent with the HRRR and RTMA deficiencies noted by
forecasters. In February 2020, NOAA changed from the RTG_SST_HR to a Near-
Surface Sea Temperature (NSST, see NWS, 2020) for SSTs, but using the same
horizontal interpolation method to estimate small-lake temperatures resulting in the
same temperature biases for small lakes.

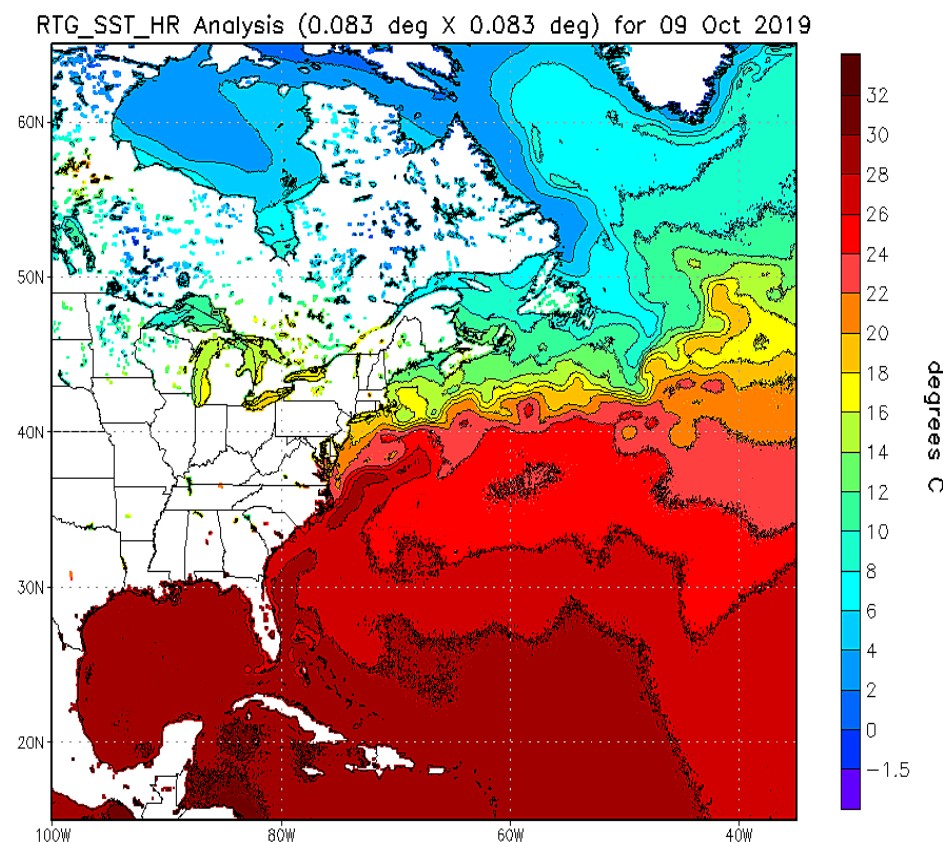

22:40:13 WED OCT 9 2019

*Fig. 3. An example of small-lake temperatures spatially interpolated from deeper-water*
*temperature data in the NOAA SST analysis (Gemmill et al, 2007). For 9 October*
*2019, provided by NOAA National Weather Service.*

Hamill (2020), in a comparison benchmarking a statistical method for 2-m temperature
(at 00 UTC), showed the same problem with large summer temperature biases from the
HRRR 1-h forecasts in August 2018 especially in the vicinity of lakes (his Figs. 10, 11).



His results are shown in Fig. 4, with three stations showing coldest biases (at 00 UTC)
greater than 2 K (circled in red), all adjacent to lakes.  In Fig. 4, these circled stations,
from north to south, are KFGN (Flag Island on Lake of the Woods; >3 K cold bias),
KRRT - Warroad, MN (west of Lake of the Woods), and KVWU – Waskish, MN (east of
Red Lake)). The overall warm or cold biases are generally < 2 K, but these stations
adjacent to lakes are outliers, consistent with introduction of cold-biased lake
temperatures through the NSST.

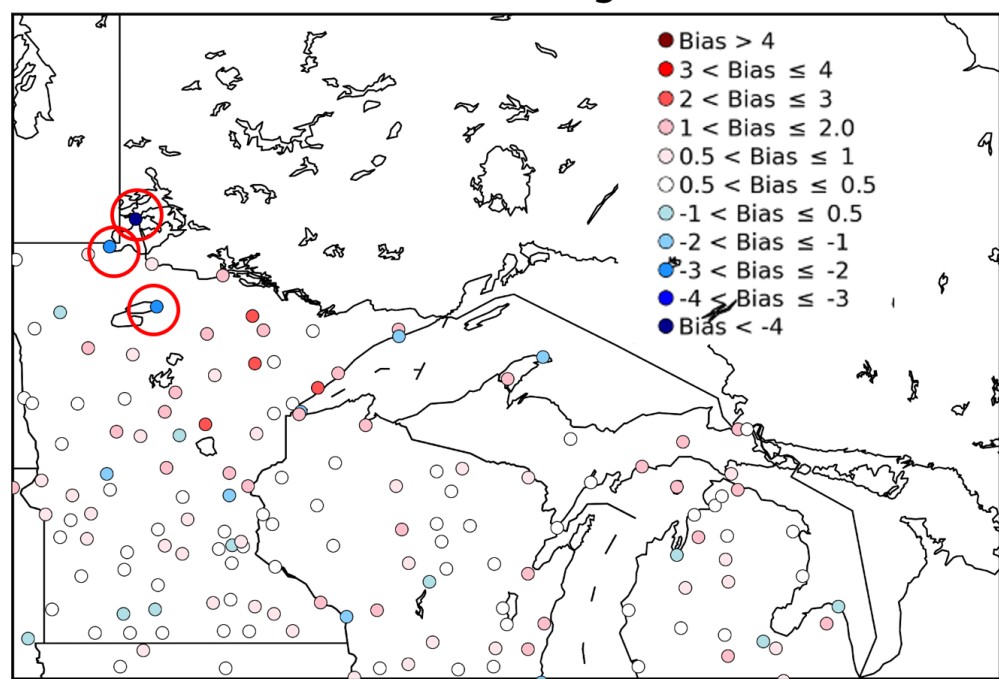

*Figure 4.  2-m temperature biases for 1-h HRRR forecasts valid at 00 UTC in August*
*2018 (from HRRRv3, before introduction of lake cycling and using NSST estimates*
*instead).  Stations with low bias < -2 K are circled in red.  (Credit and thanks to Thomas*
*Hamill, providing a regional version of his Fig. 10b in Hamill, 2020).*
With its 3-km grid spacing, the HRRR model can resolve many inland lakes (Fig. 1a).
Specification of surface temperatures for these small lakes using the horizontal
interpolation from the NOAA SST fields was problematic being determined by
interpolation from large lake and ocean temperatures.
In summary, errors in specified lake temperatures (as well as ice cover and
concentration) due to spatial interpolation from oceans and larger lakes can lead to
degraded atmospheric predictions in the vicinity of lakes. For small lakes, poor short-





range 2-m temperature (T) and 2-m dew point temperature ($T_d$) forecasts were noted in
vicinity of lakes, especially from spring through summer and into autumn. Specifically,
fluxes from lakes were often poorly estimated due to inaccurate lake temperature fields.
## 3   Lake model for coupling with NOAA regional atmospheric models
Similar to the now-commonplace (in NWP models) coupling with land-surface models
(LSMs) to improve fluxes into the atmosphere, a multi-level 1-d lake model was
implemented within the operational 3-km HRRR and 13-km RAP weather models in
December 2020, an extension to atmosphere-surface coupling. An effective lake
initialization is a necessary complement for the lake model coupling, as described in
section 4. Different earth-system coupling processes represented in the HRRR and
RAP models are described in Table 1, including land, snow, ice, and smoke. The
Community Land Model (CLM) lake model (same in versions 4.5 and 5.0) was added
for smaller lakes as an option in the WRF model version 3.6 (Mallard et al, 2015). The
CLM lake model is described in more detail below with its configuration for the NOAA
HRRR and RAP weather models.  A detailed description of the physical processes
(cloud microphysics, turbulent exchange, land-surface, etc.) in the HRRR and RAP
models are described by Dowell et al (2022) and Benjamin et al (2016).



| Component | Prognostic variables | Layers (below surface except for smoke) | Year introduced for experimental cycling | Year intro for NCEP | Data assimilation | Other information, references |
|---|---|---|---|---|---|---|
| **Soil** | Temp, moisture | 9 | 1996 (6 levels until 2012) | 1998 (6 levels until 2014) | Cycling, atmos-to-soil coupled DA | **Moderately coupled DA** (Benjamin et al 2022) |
| **Snow** | Water equiv, snow depth, temp | 2 | 1997 | 1998 | Cycling, atmos-to-snow DA for temp, trim/build from sat for cover | **Moderately coupled DA.** Subgrid fraction intro 2020 |
| **Ice** | Temp | 9 | 2010 (6 levels until 2012) | 2012 (6 levels until 2014) | Cycling, atmos-to-surface coupled DA | Subgrid fraction intro 2018 |
| **Smoke** | Smoke mixing ratio | 50 atmos layers | 2016 | 2020 | Cycling, fire rad power from sat | No direct DA, only cycling |
| **Small lakes** | Temp, ice fraction, mixing | 10 | 2018 | 2020 | Cycling | No direct DA, only cycling |
| **Large lakes (Great Lakes)** | Temp, ice fraction, mixing | FVCOM levels | 2018 | 2020 | Independent | FVCOM driven by HRRR wind, rad, temp, 6h lag (Fujisaki-Manome et al 2020) |

*Table 1.  Earth-system coupling added to NOAA regional models (HRRR, RAP, RUC*
*(pre-2012))*
An additional improvement in lake-atmosphere coupling in NOAA weather models was
recently introduced, a coupling between the NOAA HRRR model using predicted lake
temperatures and ice concentration fields from the NOAA GLERL/NOS 3-dimensional
hydrodynamic-ice model run in real time over the Laurentian Great Lakes, as described
by Fujisaki-Manome et al (2020). This hydrodynamic-ice model is based on the Finite
Volume Community Ocean Model (FVCOM, Chen et al., 2006, 2013) coupled with the
unstructured grid version of Los Alamos Sea Ice Model (CICE; Gao et al., 2011) and is
applied to the Great Lakes Operational Forecast System (GLOFS, Anderson et al.,
2018). This time-lagged data coupling (alternate applications of HRRR atmospheric
forcing and FVCOM-CICE lake forcing about 6-12 h in advance) was incorporated to
improve lake-effect snow (LES) predictions in winter but has also been found to improve
near-lake atmospheric predictions year-round especially for upwelling events in the
warm season. The use of FVCOM-CICE to specify lake temperatures addresses



previous errors in SST from relatively fast changes in lake temperatures due to cold air
outbreaks or upwelling events. These changes sometimes escape AVHRR-derived SST
detection due to multi-day cloud obscuration.

| Small lake size (grid points) | # Lakes | % of # of small lakes | % of small lake surface coverage | Avg depth (m) | Surface area of lakes (km²) | Volume of lakes (km³) |
|---|---|---|---|---|---|---|
| 1 grid point (3kmx3km) | 917 | 49% | 7% | 13 | 8,812 | 115 |
| 2 (~20 km²) | 323 | 17% | 5% | 12 | 6,208 | 76 |
| 3 | 155 | 8% | 4% | 11 | 4,468 | 49 |
| 4-5 | 157 | 8% | 6% | 14 | 6,746 | 97 |
| 6-10 (~100 km²) | 155 | 8% | 10% | 14 | 11,570 | 162 |
| 11-100 (~1000 km²) | 141 | 7% | 30% | 21 | 35,518 | 769 |
| >100 | 16 | <1% | 38% | 14 | 44,926 | 614 |
| **All** | **1864** | **100%** | **100%** | | **118,248** | **1,882** |

*Table 2.  Characteristics of small lakes (not including the five Laurentian Great Lakes)*
*resolved in the 3-km HRRR CONUS domain over the lower 48 United States and*
*adjacent areas of Canada and Mexico.  Grid points were assigned as having a lake land*
*use for points with at least 50% lake representation from the higher-resolution 15"*
*MODIS land-use data.*

| Laurentian Great Lakes | Surface area of lakes (km²) | Volume of lakes (km³) |
|---|---|---|
| **Superior** | 82,100 | 12,000 |
| **Michigan** | 57,800 | 4,920 |
| **Huron** | 59,600 | 3,540 |
| **Erie** | 25,670 | 484 |
| **Ontario** | 19,010 | 1,640 |

*Table 3.  Characteristics of the five Laurentian Great Lakes (surface area, volume)*
*(Hunter et al 2015).*



3.1  CLM lake model applied to HRRR for smaller inland lakes
Subin et al (2012) describe the 1-d CLM lake model as applied within the Community
Earth System Model (CESM) as a component of the overall CESM CLM (Lawrence et al
2019). Gu et al (2015) describe the introduction of the CLM lake model into the WRF
model and initial experiments using its 1-d solution for both Lakes Superior (average
depth of 147 m) and Erie (average depth of 19 m). The CLM lake model divides the
vertical lake profile into 10 layers driven by wind-driven eddies. The atmospheric inputs
into the model are temperature, water vapor, horizontal wind components from the
lowest atmospheric level and short-wave and longwave radiative fluxes (from the HRRR
model in this application).  The CLM lake model then provides latent heat and sensible
heat fluxes back to the HRRR. The CLM lake model is called every 20 s within the
HRRR model.  The CLM lake model was configured with the top layer fixed to a 10-cm
thickness (Gu et al 2015) and with the rest of the lake depth divided evenly into the
other 9 layers. Energy transfer (heat and kinetic energy) occurs between lake layers via
eddy and molecular diffusion as a function of the vertical temperature gradient. The
version of the CLM lake model used for HRRR and RAP was introduced with CLM
version 4.5 and continues without change in CLM version 5 (Lawrence et al, 2019). The
CLM lake model also uses a 10-layer soil model beneath the lake, a multi-layer ice
formation model and up to 5-layer snow-on-ice model (Gu et al, 2015). Testing of the
CLM lake model by the authors within WRF showed computational efficiency of the
model with no change of even 0.1% in run time with the HRRR and RAP applications.
Multiple layers in lake models better represent vertical mixing processes in the lake. By
intention, the CLM lake model was only applied for HRRR and RAP model to smaller
lakes, since NOAA began at the same time to provide temperature and ice cover
through GLOFS for the Laurentian Great Lakes through the 3-d hydrodynamic-ice
model (Fujisaki-Manome et al, 2020, Anderson et al, 2018).
3.2   Lake area mask
Grid points were assigned as lake points when the fraction of lake coverage in the grid
cell (derived from yet finer 15" MODIS data) exceeds 50% and when HRRR gridpoint
elevation > 5 m above sea level (to distinguish from ocean). The lake water mask is
therefore binary, set to either 1 or 0.  This binary approach at 3 km seemed capable of
capturing the effect of lakes on regional heat and moisture fluxes.  The alternative
subgrid lake fraction approach was used by ECMWF with their 9-km model (Choulga et
al, 2019).
An overview of the lake number, areal coverage, and integrated volume for the 3-km
HRRR model are depicted in Table 2. The HRRR CONUS domain (Fig. 1a) is able to
represent 1864 separate lakes occupying 0.6% of the entire domain. These water
bodies represented in HRRR as "lakes" include reservoirs and larger rivers, and about





half of the 1864 lakes are single-gridpoint lakes. The sixteen largest lakes in the HRRR
CONUS domain have surface area greater than 1,000 km$^2$, nine in Canada and two on
the US-Canada border (Lake of the Woods and Lake St. Clair).  In contrast, the five
Laurentian Great Lakes (Table 3) range in size from 82,000 km$^2$ (Superior) to 19,000
km$^2$ (Ontario), and therefore, their representation in the coupled HRRR system (Table 1)
is handled with 3-d hydrodynamic-ice models (Fujisaki-Manome et al, 2020).
The lake area mask for the 3-km HRRR used an algorithm for identifying an ocean area
mask for all areas with contiguous water areas and leaving other areas as near-ocean
lagoon regions treated as lakes with the CLM 1-d lake model.   These lagoon areas
separated from ocean by barrier islands in the HRRR representation (Fig. 1a) include
the Intracoastal Waterway in Texas largely separated from the Gulf of Mexico by Padre
Island, Indian River in Florida largely separated from the Atlantic Ocean by Merritt
Island, and Lake Pontchartrain in Louisiana.  This ocean-contiguity technique is similar
to the flood-filling technique used by ECMWF (Choulga et al, 2019).
3.3.  Lake depths
Lake depths for the HRRRv4-WRF-CLM lake configuration are assigned from a global
dataset provided by Kourzeneva et al (2012, hereafter K12).  For some smaller lakes
identified using the 15" MODIS land-water mask not found in K12, a 50-m depth was
assumed.  ECMWF applied a 25-m depth as a default depth for these small lakes
(Choulga et al, 2019).  For many lakes in the K12 database, a single value for maximum
lake depth had been applied to all lake points, which results in excessive lake water
volume and too cold temperatures as discussed in section 5. However, the K12
database still allows overall differentiation between shallow and deep lakes.  The
majority of the small lakes in the northern US and southern Canada are assigned as
shallow, at 5-m depth, but a few are assigned a depth as 30 m or deeper (Fig. 5).

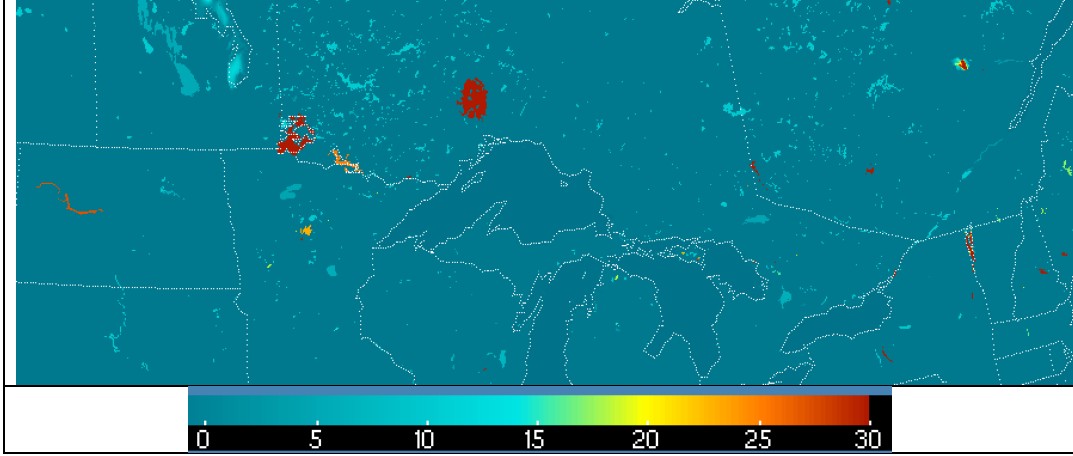

*Figure 5.  Lake depth for small lakes in a subset of the HRRR domain with red for lakes*
*30 m or deeper.*





3.4 Turbidity
A single value for turbidity to describe absorption of downward short-wave radiation is
used in CLM, allowing for a moderate amount of suspended sedimentation. Subin et al
(2012) describe other options for variations in radiative transfer in lake bodies to capture
degrees of eutrophication, but these are not used here.
3.5 Salinity
The CLM lake model is configured for fresh water. The authors manually modified the
freezing temperature to account for non-zero salinity (Railsback, 2006) from 0°C to -5°C
for Mono Lake in California and Great Salt Lake (GSL) in Utah to capture the effect of
salinity.  Other areas of water impoundment from coastal lagoons in the 3-km HRRR
lake representation (Fig. 1a) also have non-zero salinity (e.g., along coasts of Gulf of
Mexico and Atlantic Ocean) but no change in freezing temperature is necessary for
these areas.
3.6  Elevation
The elevation value (above sea level) assigned to each lake grid point is the same
assigned to that from the atmospheric model, which may be different from reality, but at
least consistent with the atmospheric conditions. As mentioned earlier, the minimum
elevation above sea level of a grid point to be assigned as a lake is 5 m; other water
grid points are assumed to be ocean.
3.7  Special situations for CLM lake model application
The algorithm for the turbulent heat flux calculation in the CLM-lake model was mainly
based on Zenget al. (1998), except that roughness length scales for temperature and
humidity are the same as roughness length scale for momentum for its WRF-lake
application, while they are updated dynamically in CLM 4.5. Charusombat et al (2018)
showed that the same roughness length scales for temperature and salinity as that for
momentum could result in overestimated surface sensible and latent heat fluxes in
autumn and winter. Therefore, a revision to the CLMv4.5 lake model was introduced for
modified roughness lengths over water using modified formulations of the Coupled
Ocean-Atmosphere Response Experiment (COARE) algorithm as described by
Charusombat et al (2018) to improve surface sensible and latent heat fluxes.
For GSL with a very high value of salinity (270 ppt north of ~41.22°N with freezing point
of 249 K and 150 ppt south of ~41.22°N with freezing point at 263 K), a change of
freezing temperature to -5°C appeared to be not sufficient to keep the lake ice-free
during the cold outbreaks in winter in this high-elevation area. GSL is unusual in various





aspects – it is hypersaline (far more saline than the ocean), the largest terminal lake
(without outflow) in the Western Hemisphere (Belovsky et al, 2011), shallow (mean
depth of 5 m) and subject to very strong eutrophication (Belovsky et al, 2011).
According to GSL climatology the lake stays ice-free all winter, and its temperature goes
slightly below freezing only for a very short period in January and February. Thus, we
presume that the CLM lake model needs to allow turbidity variation (see section 3.4). A
solution to this representation problem was use of a bi-weekly climatology over each 1-
year period to bound the cycled GSL temperature at initial forecast time not to deviate
more than +/- 3°C from the climatological value interpolated to the current day of year.
Also, using special code, GSL was forced stay ice-free for the whole year as observed.
3.8  Time step
The CLM lake model within the HRRR/RAP weather models was run with the same time
step as for other physical processes in the HRRR model (20 s) and the RAP model (60
s).  Again, even with this relatively high frequency for calling the CLM lake model, the
computational expense was extremely small, less than 0.1% of overall HRRR run time.
**4   Initialization for small lake temps by cycling with defined atmospheric**
**conditions – a strategy**
The central strategy described in this paper is to use accurate, ongoing atmospheric
forcing with a computationally inexpensive 1-d lake model to obtain an equilibrium state
of a lake temperature profile.  This technique responds appropriately to strong changes
in atmospheric forcing (e.g., cold air outbreak or excessive heat events).   With the
NOAA HRRR and RAP atmospheric models performing hourly data assimilation of a
broad set of hourly observations, accurate atmospheric forcing is available.
The RAP and HRRR hourly data assimilation cycles include these aspects, all of which
are important for cycling initialization of inland lakes. First, cloud assimilation (from
satellite and ceilometer data) to ensure accurate shortwave and longwave radiation
fields (Benjamin et al 2021).  Second, radar reflectivity data are assimilated as part of a
3-km ensemble data assimilation system to ensure accurate short-range precipitation
(Weygandt et al, 2022, Dowell et al, 2022, James et al, 2022, Benjamin et al, 2016).
Finally, 2-m air temperature and moisture and 10-m wind observations are effectively
assimilated (i.e., producing more accurate predictions) including representation through
the boundary layer using pseudo-innovations (James and Benjamin, 2017).  Other
information on the HRRR/RAP data assimilation is provided by Benjamin et al (2016).
The cycling of the 10-level CLM lake model within the experimental HRRRv4 started on
24 August 2018.  After 10 days of cycling (Fig. 5), differences in lake temperatures
between HRRRv4 and the operational HRRRv3 using interpolated NSST data were

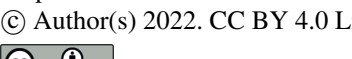



evident of 5-15°F (3-12°C or 276-285 K), showing that the adjustment with realistic
atmospheric conditions and use of the CLM lake model with roughly accurate lake depth
data was very effective.

| Consequences (to right) from strategy for lake initialization (below) | Coupling lake and atmosphere within initialization | Lake temps in spring-summer | Lake temps in fall |
|---|---|---|---|
| **SST interpolation to small lakes** | None | Much too cold, especially for shallow lakes | Still generally too cold but intermittently too warm after cold-air outbreaks. |
| **Lake annual variation forced by reanalysis atmospheric data – 1-way cycling from atmospheric forcing** | 1-way | More accurate, possibly still too cold. No regime variation in a given year | More accurate, still too cold for some lakes with too-deep bathymetry data. Will not capture variation from weather regimes in a given year |
| **2-way cycling** | 2-way | More accurate including yearly anomalies | More accurate including yearly anomalies |


*Table 4. Expected seasonal lake-atmosphere temperature consequences from different*
*lake initialization strategies*
Possible approaches for initializing lake temperatures are summarized in Table 4. The
simplest option is via larger-scale water temperature data (SST data) with horizontal
interpolation to smaller water areas including inland lakes and reservoirs; this was the
previous strategy for the HRRR and RAP models before introduction of cycling using
the CLM lake model. An alternate strategy is to run lake models over a multi-year period
forced by reanalysis atmospheric data (ERA-Interim) as described by Balsamo et al
(2012), Dutra et al (2010), and Balsamo (2013) for the ECMWF to obtain a yearly
varying climatology of lake temperature for all lakes represented. This method will
capture the mean annual variation of lake temperatures. However, due to multi-year
averaging, it cannot represent anomalous conditions in a given year (sustained heat or
sustained cold conditions), which can modify temperatures especially for shallow lakes
by several K within 1-2 weeks. Full cycling of the lake model within an ongoing coupled
weather model, the strategy described in this paper, can represent the lingering effects
of anomalously warm or cold weather upon lake temperatures and the resultant fluxes.
ECMWF applies a similar ongoing cycling for lake prognostic variables (ECMWF 2020)
for lake initialization.

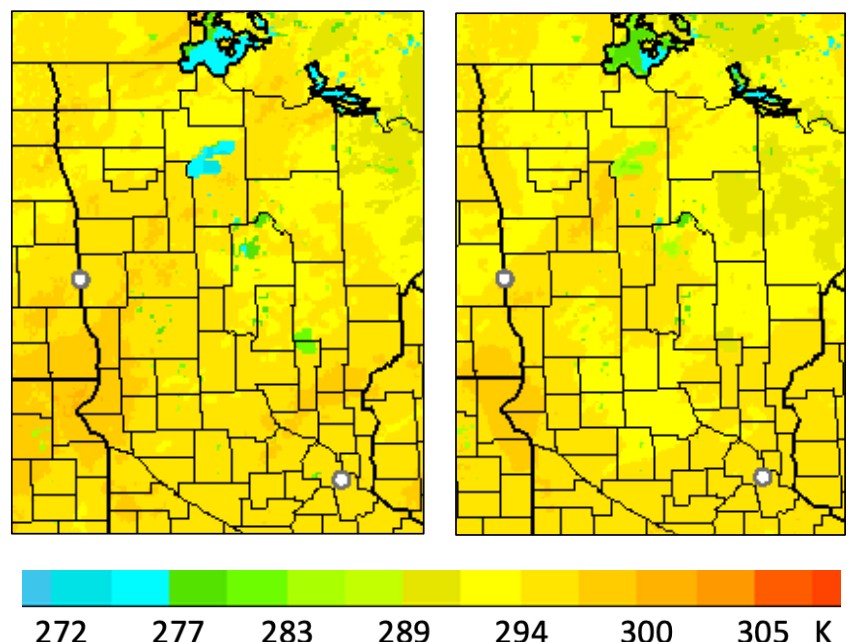

*Figure 6.  Skin temperature (K) including lake temperatures.  From 18-h forecasts valid*
*at 15 UTC 3 September 2018 for a) operational HRRRv3 using NSST for lake*
*temperatures, and b) experimental HRRRv4 with CLM lake model and cycling.*
A similar challenge is initialization of lake ice cover. Similar to the treatment for lake
temperature, cycling of a multi-level lake model (like the CLM lake model) can provide
an alternative, adaptive-in-time method for lake-ice initialization.  NOAA has used in the
HRRR and RAP the daily IMS ice cover product[1] (US National Ice Center, 2008) for
binary (non-fractional) lake ice cover. The IMS ice cover is used for oceans and large
lakes (e.g., for RAP in Fig. 1b, for Great Slave Lake and Great Bear Lake in northern
Canada). For small lakes below the resolution of the IMS ice map, lakes stayed open for
the winter before introduction of the CLM lake model with lake cycling (for grid-point-
specific temperature and ice cover) starting with HRRRv4 and RAPv5.
**5  Results**
In this section, we describe comparisons of lake surface temperature evolution between
the CLM implementation described here and the lake specification through interpolation
from the NSST dataset (Fig. 3) at lakes in the United States and southern Canada.

---

[1] https://usicecenter.gov/Products/ImsHome



Comparisons during 2018–2019 were drawn from real-time simulations from the then-
operational HRRRv3 (using interpolated SST) and the experimental HRRRv4 (using
CLM). More recent comparisons were made for March–November 2021 between the
operational HRRRv4 (using CLM) and interpolated NSST values (as used in 2019-2020
for HRRRv3).  In addition, the CLM and NSST values were compared to in situ
observations where available and also to satellite-based estimates defined below.

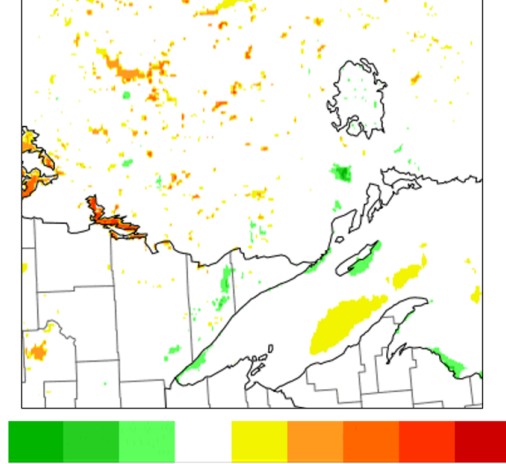

*Fig. 7. Difference (K) in skin temperature (including lake temperatures) between*
*versions of HRRR model using cycled lake-model values (HRRRv4 or HRRRX) and*
*using interpolated NSST data (HRRRv3 or HRRR-NCEP).   Valid 1300 UTC 13 October*
*2019, and also includes differences from use of FVCOM lake model in HRRRv4*
*(Fujisaki-Manome et al, 2020).*
5.1    Cases from 2018 – 2019
Introduction of the CLM lake model forced by ongoing HRRRv4 atmospheric conditions
(i.e., cycling) allowed, within only 10 days, an increase in lake temperatures for Red
Lake and Lake of the Woods (both in Minnesota) from 3 K to over 10 K (Fig. 6) in
September 2018. A comparison in skin temperature for a year later (October 2019)
between versions of the HRRR model (HRRRv4 with lake cycling vs. HRRRv3)
including differences from with and without lake cycling is shown in Fig. 7. Higher
temperatures were evident for the Minnesota/Ontario lakes from cycling (vs. NSST
interpolation).    HRRRv4 also included coupling with the 3-d FVCOM lake model for
the Laurentian Great Lakes, showing areas of upwelling with associated cooler water



over Lake Superior in Fig. 7 from predominant westerly to southwesterly near-surface
wind at this time.

| Lake number | Lake name | State/province, country | HRRR I point | HRRR j point | Area (km²) | Depth used (m) | Ice free? |
|---|---|---|---|---|---|---|---|
| 1 | Simcoe | ON, CA | 1378 | 799 | | 6 | N |
| 2 | St. Clair | ON/MI, CA/US | 1302 | 709 | 1240 | 6 | N |
| 3 | Champlain | VT/NY, US | 1534 | 835 | | 77 | N |
| 4 | Sebago | ME, US | 1610 | 833 | | 33 | N |
| 5 | Okefenokee | FL, US | 1459 | 145 | 1510 | 3 | Yes |
| 6 | Pontchartrain | LA, US | 1136 | 224 | 2180 | 10 | Yes |
| 7 | Intracoastal Waterway (near Corpus Christi, TX) | TX, US | 905 | 128 | 3300 | 10 | Yes |
| 8 | Salton Sea | CA, US | 337 | 387 | | 9 | Yes |
| 9 | Tahoe | NV/CA, US | 259 | 628 | | 313 | N |
| 10 | Great Salt | UT, US | 486 | 653 | 3050 | 3 | Yes |
| 11 | Utah | UT, US | 496 | 622 | | 3 | N |
| 12 | Bear | ID/UT, US | 518 | 684 | | 29 | N |
| 13 | Sakakawea | ND, US | 790 | 868 | | 27 | N |
| 14 | Winnebago | WI, US | 1143 | 742 | | 7 | N |
| 15 | Lower Red | MN, US | 961 | 880 | | 5 | N |
| 16 | Lake of the Woods | MB/MN, CA/US | 965 | 919 | 3030 | 32 | N |
| 17 | Manitoba | MB, CA | 879 | 972 | 3240 | 5 | N |
| 18 | Winnipeg | MB, CA | 916 | 977 | 13270 | 8 | N |
| 19 | Nipigon | ON, CA | 956 | 956 | 5410 | 55 | N |

*Table 5.  Lakes for comparison of lake temperatures between HRRR/CLM, NASA*
*SPoRT, NSST, and in situ observations as shown in Figs. 8 and 9.  Area is shown for*
*lakes >1000 km². Lake depths are constant within each lake except for lakes 2, 3, and*
*18.   See Fig. 5 for example map of lake depth used in HRRR. Specific HRRR i/j 3-km*
*grid points (indicated in table) were selected from HRRR data for each lake.*






| Name of Lake | No. from Tab. 5 | Source of Observation | Depth of Sensor (m) | URL |
|---|---|---|---|---|
| Lake St. Clair | 2 | ECCC | 6 | https://www.ndbc.noaa.gov/station_page.php?station=45147 |
| Lake Champlain - Schuyler Reef | 3 | GLERL | 0.45 | https://www.ndbc.noaa.gov/station_page.php?station=45195 |
| Sebago Lake @ Lower | 4 | Portland Water District Buoy | Est 1 | https://www.pwd.org/sebago-lake-monitoring-buoy |
| Lake Pontchartrain @ New Canal Station | 6 | NOAA/ National Ocean Service | 0.6 | https://www.ndbc.noaa.gov/station_page.php?station=nwcl1 |
| Intracoastal Waterway @ Baffin Bay near Padre Island | 7 | Texas Coastal Ocean Observing Network | unknown | https://www.ndbc.noaa.gov/station_page.php?station=babt2 |
| Lake Tahoe | 9 | NASA/JPL | 0.5 | https://laketahoe.jpl.nasa.gov/get_imp_weather |
| Utah Lake @ Provo Marina | 11 | Utah DWQ Water Quality Network | unknown | https://wqdatalive.com/public/669 |
| Bear Lake | 12 | Utah DNR State Parks | unknown | https://stateparks.utah.gov/parks/bear-lake/current-conditions/ |
| Lake Sakakawea @ Missouri River near Williston, ND | 13 | USGS | unknown | https://waterdata.usgs.gov/monitoring-location/06330000/#parameterCode=00065&period=P7D |

*Table 6.  Sources of available in situ data among 19 lakes in Table 5.*



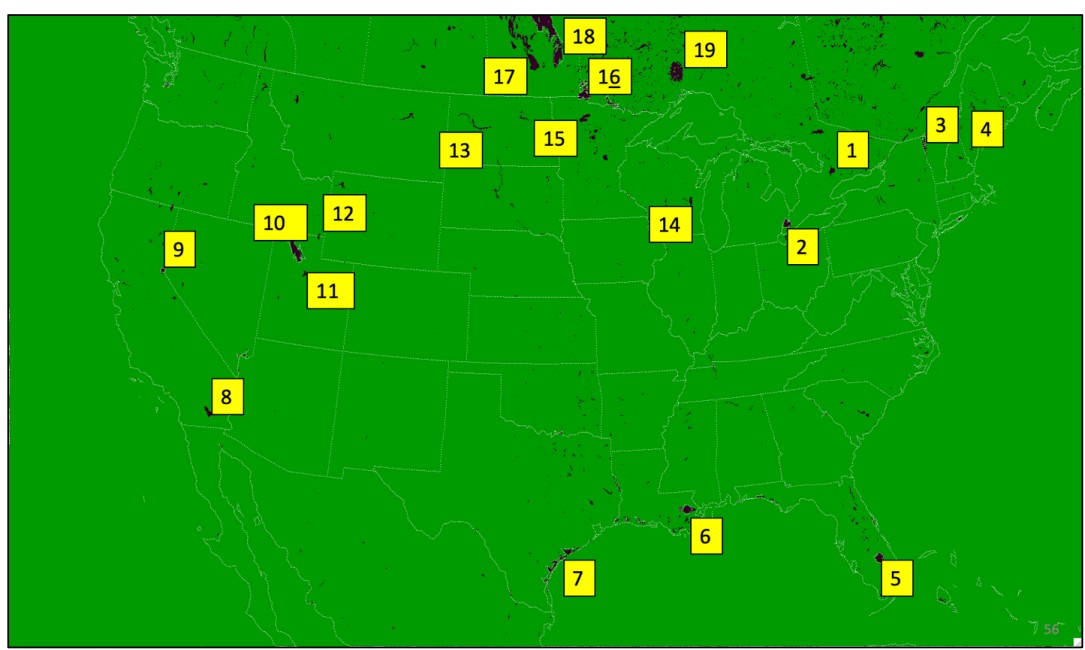

*Figure 8. Locations of 19 lakes (see Table 5) for lake temperature intercomparison.*
*These lakes are shown as mapped onto the 3-km CONUS HRRR model domain.*

5.2     Comparisons of different lake temperature estimates for 19 lakes from lower 48
       US and southern Canada during 2021.

During a period from March to November 2021, a comparison was made of lake
temperatures between the cycled HRRR-CLM values and those from three other
estimates from NASA, NOAA, and in situ observations. A geographically diverse set of
19 lakes over the lower 48 United States and southern Canada was selected for these
comparisons as listed in Table 5 and shown in Fig. 8. Lakes selected included near-
ocean lagoon areas separated from ocean areas by coastal land as resolved by the 3-
km land-water mask as discussed in section 3.2. The water areas also included a
reservoir (Lake Sakakawea). Some of these lakes are dimictic or polymictic (with ice
cover part of each year, Lewis 1983) but five of them do not experience any ice cover
(Table 5), and lakes 5, 6, 7, and 8 are monomictic.  The CLM lake model was cycled for
all these lakes in the 3-km HRRR model. The 19 lakes included seven lakes with a
surface area greater than 1,000 km². The March-November evaluation period include
the spring-summer warming period and the cooling period in autumn. Data points were
obtained monthly for March-August and weekly for September-November.





541







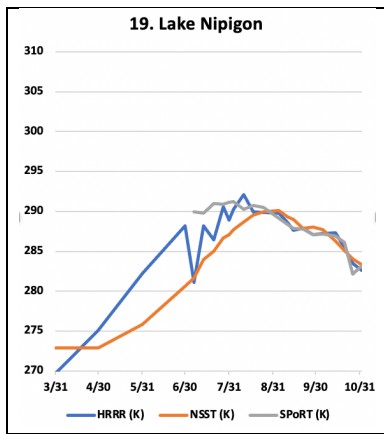

*Figure 9.  Lake temperatures in 2021 (April-October) from the 19 selected lakes (Table 5, Fig. 8) from HRRR-CLM-cycled (blue), NSST (red), SPoRT (gray), in situ (orange).*

The HRRR-CLM values for these 19 lakes were compared with first, an estimate from NASA SPoRT (Short-Term Prediction Research and Transition) real-time surface water temperature composite including time-weighted MODIS and VIIRS data for inland lakes (NASA, 2021, Kelley et al, 2021). The composite is valid from the surface to 2-m depth and is averaged over a 7-day period to mitigate for cloud cover on a given day. A second lake temperature estimate is that from NSST, as discussed earlier. Third, in situ surface water temperature observations were available from observing platforms in nine of the 19 lakes (Table 6).   The platforms are operated by Federal, state, and local government agencies and a regional ocean observing system. The depths of the water temperature observations were only available at four of the nine platforms. At these four sites, the depth ranged from 0.45 to 0.9 m.

In general, the HRRR-CLM-cycled lake temperatures showed the anticipated difference from NSST values with quicker summer warming from HRRR-CLM cycling for all lakes except the southern 3 lakes (5, 6, 7 in Table 5, with Lakes 6 and 7 essentially lagoons in close proximity to the ocean) and Bear Lake in UT/ID (Lake 12, 39-m depth). The NSST estimates were colder for spring through summer than HRRR values for 15 of the 19 lakes, a consequence from the NSST estimate via horizontal interpolation from deeper bodies of water.

For the nine lakes with in situ observations (Table 6), the HRRR-CLM-cycled lake temperatures are generally able to better capture weekly variability in summer and autumn months, associated with windy periods increasing mixing or relatively warm and cool weather periods or varying amounts of cloud cover.  This can be seen, for example, at Utah Lake and the Intracoastal Waterway west of Padre Island in Texas (note cooling from passage of Hurricane Nicholas in mid-September).  The most


dramatic improvement of HRRR-CLM over NSST lake temperatures is seen at Lake
Tahoe and lakes 14-19 in the northern region, with NSST estimates 5-10 K too cool.  At
two of the lakes with in situ observations, the Intracoastal Waterway (linked to the
ocean) and Lake Pontchartrain, both lagoons linked to the ocean, NSST estimates are
generally closer than HRRR-CLM to the observations.
HRRR-CLM lake temperatures matched in situ observations well for the northern lakes,
usually within 1-2 K.   In contrast, the lake temperature values from SPoRT were
generally warmer than HRRR or in situ observations in the autumn period.  The SPoRT
observations showed a strong confirmation of HRRR-CLM-cycled lake temperatures for
lakes in the western US (Lakes 8-13) and most lakes in the northern areas (Lakes 4,
14-19).  Finally, the HRRR-CLM-cycled lake temperatures during this period often
varied strongly from the NSST estimates, with differences of up to 5-10 K (largest
difference with Red Lake,  Lake 15).   The effect of lake depth was evident with a faster
transition to fully mixed lakes for shallow lakes (e.g., 5-m depth for Red Lake in MN,
Lake 15 in Table 5) but subject to more temporal and horizontal variation for deeper
lakes.   Fig. 10 showed a strong intralake variation of 9 K across Lake of the Woods
(32-m depth) in the HRRR-CLM estimate in contrast with very little variation (< 1 K)
across Red Lake.  Due to a lack of high-resolution observations of lake surface
temperatures, it is difficult to determine which intralake variations are more realistic.
However, we think some of these intralake contrasts from HRRR-CLM may be
exaggerated from actual values, possibly requiring introduction of a small temperature
exchange rate (diffusion) between adjacent lake columns.  Differences in skin
temperature (e.g., SPoRT) and bulk temperature (e.g., in situ) for lakes have been
noted (e.g., Wilson et al, 2013) of up to 0.5 K, but the HRRR vs. NSST differences in
this study are generally much larger than this magnitude.


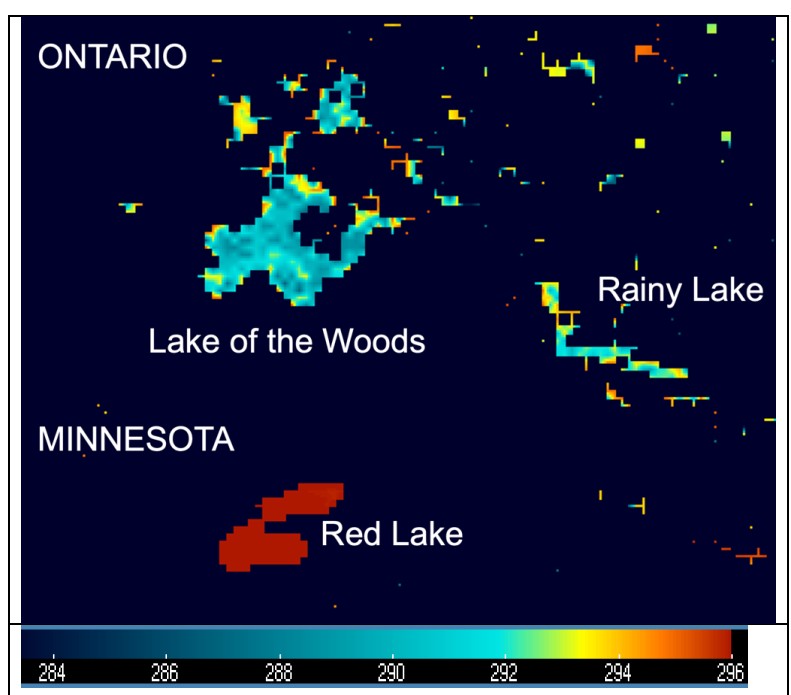

Fig. 10.  *HRRR-CLM lake temperature (K) for 1500 UTC 31 July 2021 for area over*
*northern Minnesota (US) and southwestern Ontario (Canada).*

The main deficiencies evident so far with the HRRR-CLM lake temperatures appear to
be associated with errors in lake depth values. On the average, the lake depth for most
lakes is too deep, since the preprocessing with the K12 dataset simply assigned a
single lake depth value (maximum or mean) to all grid points for that lake even up to the
modeled lake points adjacent to land, as shown in Table 5 for 16 or the 19 lakes
studied. We also noted too-low lake temperatures in HRRRv4 for lake grid points at the
western edge of a few lakes (e.g., Tahoe, Sebago (ME), Cayuga (NY), Champlain), all
relatively deep lakes (Fig. 6, Table 5).  We attribute this to 1-d upwelling from
insufficient bathymetry data resulting in cylinder-like lake volumes with constant lake
depths, therefore with a) too-deep lake-edge pixels coinciding with b) strong winds
coming off from land areas with predominantly westerly winds.  This deficient effect was
not widespread for the HRRR model and did not affect the overall results. Again, this
behavior is attributed to the behavior of the lake model over integrations with the
inaccurate lake depth information and not to the lake cycling initialization design.

**6   Conclusions**

We report here on the first use of a small-lake model (CLM4.5, 10 layer) in US NOAA
NWP models along with an ongoing cycling of lake temperatures since 2018 to initialize





lake temperatures in each prediction. These models are the 3-km HRRRv4 (D22, J22)
and 13-km RAPv5 hourly updated models, both of which became operational in
December 2020 after cycling since August 2018. At 3-km grid spacing, the HRRR
model applied this small-lake modeling and assimilation to 1864 small lakes varying in
size from about 10 km$^2$ (single grid point) to 14 larger lakes over 1000 km$^2$ in surface
area, but not including the Laurentian Great Lakes. The effectiveness of introducing the
multi-layer lake model into the HRRR and RAP models was completely dependent on
the initialization for lake temperatures. The introduction of a cycling capability through
the hourly assimilation allowed the lake temperatures to evolve to accurate values,
consistent with recent weather. In this paper, we describe the lake cycling applied for
the NOAA regional 3-km HRRR and 13-km RAP weather models including the coupled
1-d CLM lake model. We also show some comparisons with other estimates of lake
temperatures. From those comparisons, the cycled lake temperatures from the 3-km
HRRR model were found to be reasonably accurate. HRRR lake temperatures were
found to be generally within 1 K of in situ observations and within 2 K of the SPoRT
estimates. Finally, NSST estimates of small-lake temperatures were found to often differ
from in situ observations and HRRR estimates by 5-12 K. Other differences between
lake-cycled HRRR estimates and SST-based estimates were up to 10-15 K.
From these initial results, we conclude that the lake-cycling initialization for small lakes
has been effective overall, owing to accurate hourly estimates of near-surface
temperature, moisture and winds, and shortwave and longwave estimates provided to
the 1-d CLM lake model every time step (20 s for 3-km HRRR model). The HRRR-CLM
treatment also allows some inland lakes to freeze in winter, which is more consistent
with observations.  The lake cycling strategy is similar to that initialization method used
by ECMWF for its 9-km (as of 2021) IFS (Integrated Forecast System) and using a
binary lake mask in the 3-km HRRR model.
One deficiency noted due was development of too-cold lake surface for a few lakes on
their western boundary.  We attribute this to the incorrect bathymetry data with constant
lake depth (e.g., see caption for Table 5**)** causing an excessive 1-d upwelling from too-
deep lake depth at western shores for these lakes. This issue is being addressed with a
current project to improve lake bathymetry data for which results will be reported in the
future.  Also, HRRR-CLM cycling gave poorer results than NSST at least for Lake
Pontchartrain (Lake #6 in Table 5), suggesting to use NSST for near-ocean lagoon
areas.   More investigation is needed for strong intralake variations overall in HRRR-
CLM-cycling representation (e.g., Lake of the Woods in Fig. 10) and possible
introduction of horizontal diffusion of temperature between adjacent lake points.
US NWS forecasters have reported much improved near-surface temperature and
dewpoint predictions in the vicinity of small lakes from the 3-km HRRR model in 2021
since the implementation of the 1-d CLM lake model and lake-cycling initialization.
Again, this effort complements the coupling of the HRRR model with the 3-d FVCOM
hydrodynamical lake model for the Laurentian Great Lakes (Fukisaki-Manome et al,



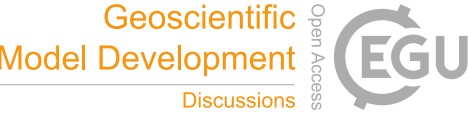

2020) design to improve lake-effect snow predictions. These efforts are the most
advanced lake-coupling and lake-initialization efforts at this point in US NOAA weather
models.
Overall, the improved lake temperatures from the lake cycling initialization technique
driven over a 3-year period by accurate atmospheric conditions described here results
in improved fluxes of heat and moisture over using SST interpolation and improved
nearby predictions of atmospheric 2-m temperature and 2-m moisture.

**Code availability**

This research used WRF version 3.9.1 including use of the option with the CLM lake
model.  All code is available from the National Center for Atmospheric Research
(NCAR) at https://www2.mmm.ucar.edu/wrf/users/download/get_sources.html

**Data availability**
HRRR data are publicly available via archives hosted by Amazon Web Services
(https://registry.opendata.aws/noaa-hrrr-pds/) and Google Cloud Platform
(https://console.cloud.google.com/marketplace/product/noaa-public/hrrr?project=python-
232920&pli=1).

**Author contributions**

SB, TS, and EJ planned the design.   TS and EJ carried out the actual coding for
modeling, data assimilation and scripts.  EJ, SB, JK, and SK extracted data from
experiments and other sources.   EJ and JK analyzed the results.   SB wrote the
manuscript draft and led its revision.  EA, AFM, JK, GM, AG and PC (along with TS and
EJ) reviewed and edited the manuscript.

**Acknowledgments**
Credit is due to the WRF model team at NCAR (Jimy Dudhia) for their help in applying
the CLM lake model for the HRRR and RAP applications of the WRF model.   We
greatly appreciate our NOAA colleague, Thomas Hamill (NOAA PSL), for Fig. 4 from
another already published article by him.  We also thank Frank J. LaFontaine and Kevin
K. Fuell of the NASA SPoRT Team for providing archived Northern Hemisphere SST
composites.  Thanks also to Rob Cifelli of NOAA/PSL for a very helpful review of our
manuscript.  This work was supported by NOAA Research base funding.

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
