# Peer review of "Inland lake temperature initialization via cycling with atmospheric data assimilation"

_Geoscientific Model Development, 2021_

## Author Response (AR1)

**GMD-2021-409 response to Anonymous Reviewer #2  - 21 July 2021**

**Thanks to Anonymous Reviewer #2 for her/his very helpful comments. Responses are shown below in bold font.**

Overall this paper provides and important and interesting contribution to operational coupling of lake temperature and weather models, I enjoyed reading the paper and the structure of the arguments. Fundamentally the science presented here should be published and is appropriate for the journal audience, however the manuscript needs additional work in two significant areas.

Firstly, a more comprehensive methods description is required describing how the assimilation was implemented, and secondly the figures could be significantly improved (there is much repetition and little content on most map based figures). I also found that the early context setting in the introduction assumes a lot of prior knowledge and believe the article could be made more accessible with some relatively minor alterations to this section.

**These are excellent suggestions and we agree with them.**

My specific comments are below.

Line 71: Could you define SST on first use.

**Done**.

Line 71-76: I would like a more specific background here on what is done operationally (e.g. which models and data) and how that differs between the great lakes and smaller non-great lakes. I think this makes too many assumptions about how well the reader will know the problem being addressed and there are not even any references in this section which should form a key component of the study rational. I appreciate a more comprehensive review is provided later, but I think this section needs to set out the problem better.

**We have revised this paragraph and agree that it needed to be much clearer.   Here is our revised paragraph:**

*In operational US NOAA weather prediction models (global and regional) up to this point, daily sea-surface temperature (SST) analyses have been used to specify the surface water temperatures for even small inland lakes.  Inland lake temperatures in North America have been obtained by the interpolation of SST values from the ocean and the Laurentian Great Lakes.   An alternative is to incorporate one-dimensional (1-d) lake models within NWP models and use coupled cycling forced by atmospheric conditions updated by new observations*

*and continuously simulated 1-d lake models   to obtain realistic lake water temperatures (e.g., "cycling").  This cycling to initialize small lakes in NOAA operational regional weather prediction models complements loose coupling with a 3-d hydrodynamical lake model for the Laurentian Great Lakes as described elsewhere in Fujisaki-Manome et al 2020.*

Figure 1a/b, It's quite difficult to see the lakes in these figures. Could you provide an insert map to zoom in on an example area so that the reader gets a better impression of the data.

**We have merged Figs. 1a and 2 into a single figure now with an insert zoom map for a region near the state of Wisconsin.  We also decided to delete Fig. 1b since the emphasis in the paper is for the 3-km HRRR model even though the same lake initialization is still used for the 13-km RAP model.**

Could figures 1a and 2 be combined? And perhaps you could label the lakes to link with Table 5? I'm not convinced of the need for Figure 2 and 8, perhaps these become too busy if combined but a more efficient use of plots seems possible.

**After merging Figs. 1a and 2 and adding an insert, we felt it would be best to keep Fig. 8 separate.  There is still a reduction of 2 graphics (Fig. 2 and Fig. 1b) overall.**

Line 317: Is this ocean separation different to the elevation thresholding method described at line 300?

**This is also an excellent point and our paper was confusing on this point.  Yes, there are 2 factors to identify lagoon areas: <5m ASL and disconnected from ocean points using the 3-km land-water mask.  A revised first sentence in section 3.2 now reads*:
Grid points were assigned as lake points when the fraction of lake coverage in the grid cell (derived from yet finer 15" MODIS data) exceeds 50% and when HRRR gridpoint elevation > 5 m above sea level (ASL, to distinguish from ocean) and is disconnected from ocean areas with the 3-km land-water mask.*

**Additional wording was also added later with the discussion of lagoon identification to combine these 2 factors.**

Line 329: Global lake products will include significant uncertainties, could you briefly outline what is known about these here?

**We added a new sentence here to address this uncertainty:   *K12 identified uncertainties in their own database including estimates of lake depth and errors in coastlines*.**

Line 449: I was expecting to see a description of the assimilation method. What is provided here is too brief given the focus of the paper and journal style. If these details really mess up the flow of the article this methodological detail could take the form an appendix.

**We agree – the flow of the text promised some discussion on the actual assimilation but there was nothing in the previous version.   We now have a new paragraph reading as this:**  *The 2-way cycling (Table 4) used now in the HRRR and RAP models benefit via hourly data assimilation using latest hourly observations both for the atmosphere (D22) and land-surface snow conditions (Benjamin et al 2021). In the 3-km HRRR model, the 3-d state of the atmosphere, land surface, and inland lake conditions are advanced on 20-second time steps using the HRRR-specific configuration (described in D22) of the WRF model (Powers et al, 2017; Mallard et al, 2015).   As atmospheric conditions change every 20 s (including temperature, moisture, wind, and radiation), the exchange of heat, moisture, and momentum between inland lake points and the atmosphere also vary.  Lake temperature is not modified in the hourly data assimilation step, but the ongoing exchange recalculated every 20 s forces an evolution of lake conditions to values consistent with atmospheric conditions.*

---

## Author Response (AR2)

**GMD-2021-409 response to Anonymous Reviewer #1  - 28 July 2022**

We now submit a new version of our manuscript for consideration by the reviewer and the editor.    We appreciate your challenge to bring our manuscript to another level that will seem appropriate for the GMD special issue on modeling inland lakes.   In particular, we have remade all of the figures except for two that were already more suitable for a professional journal.   We interpreted the reviewer statement on the manuscript seeming like a 'report' rather than a journal article was driven especially by the non-professional quality of the figures in our previous submission. The manuscript already contains sufficient details that allow the reproducibility (provided that one has sufficient computing resources and labor for a large operational system like the one in this study), including those for the atmospheric model (WRF) and land lake model (CLM), lake representation (mask, depth), parameterizations for key processes (turbidity, heat fluxes), and data assimilation. The public repositories for accessing the codes and data are listed in the code and data availability statements.

A response was already posted on 19 May 2022 to the reviewer's initial review.  We add some information below.  We also note that the latest version also includes many changes made in response to Anonymous Reviewer #2 on 16 June 2022.

1. It is very difficult to follow the scientific content of the paper. The manuscript seems to be very well suited for an internal report - where readers already know about the models details -  and less to a scientific manuscript.  The manuscript has to be profondly revised. I do not see how other research group can benefit from this study with the current layout. I need much more technical details for a paper in GMD. Note that I am not questionning the quality of the work here.

The unique aspect of our paper is the data assimilation used to provide improved initial conditions for lake model when coupled to a weather model.   This technique is unique in North America, and our paper is to provide evidence that this data assimilation method works very well,   We are not aware of other papers that provide such a comparison for the lake initialization methods.

2. litterature review. The litterature review missed many important contributions on the two way coupling lake atmosphere exchanges. I have added a non exhaustive list: I was surprised to not see references to COSMO/FLAKE (http://www.borenv.net/BER/archive/pdfs/ber15/ber15-218.pdf  , **http://www.cosmo-model.org/content/model/modules/flake/**), Simstrat (https://doi.org/10.1038/s41598-021-04061-6),  CRCM (https://doi.org/10.1080/07055900.2000.9649657 ) etc

Additional references were added in our previous revision.

3. Figures. I do not see the added values of most figures showing maps of North America. Figures looks more like print screens than carefully designed visual information

The latest version of our manuscript is improved in particular in this regard with much improved figures. The maps are necessary to provide information on the specific testing for the 19 lakes as described in Fig. 8 and the associated text.

4. L37 "errors in lake temperature from as much as 5-10K" I am not aware of any model with such range of error. This error range does not make sense.

The results in Fig. 8 show that indeed the use of the interpolated SST data can result in such errors.

5. L86 "However, lake temperature initialization is still a problem." It is not clear why it is a problem. 1-D models are fast to run and can easily be run for long period with no memory from the initial conditions.

The results of our paper demonstrate that the alternative method used by the previous US NOAA models have provided poor lake temperatures. The new data assimilation by continuous simulation with updated atmospheric conditions (hourly for the HRRR model) result in an improved result, as shown in Fig. 8.

6. I question the reproducibility of this study. The authors do not provide their codes/working examples. Again, I do not see how other research group can benefit from this study. This study is not FAIR-compliant and do not make a contribution valid for GMD in the present form

We again underscore that the new data assimilation method is the unique aspect of our manuscript, not a change to the lake model. We think that we have been successful in demonstrating its success.

Below is a summary of the changes made in this version.

Fig.1. In this figure, as in many of the others, a white background is used. The overall quality is much sharper. The areas of actual lake cover are somewhat more evident than in the previous version.

Fig. 4. Topic: Sample of lake depth data resolved by the 3-km HRRR weather prediction model. Again, a similar new white background is used with much sharper.

Fig. 5. Topic: comparison of lake surface temperatures.   In this new version, we show only the lake temperature data and have removed all of the distracting soil temperature data.   Again, the background is now white.

Fig. 6.   Similar changes as in Fig. 5**.**

Fig. 7.  Similar changes as in Fig. 1.

Fig. 8.  Completely redone to represent the lake surface temperatures trends into a single-page figure.   This figure is the most central result of our paper and represents evidence that our continuous lake simulation with constantly updated atmospheric conditions (e.g., "cycling") is effective.

Fig. 9.  Again, a white background is now used.   The representation of the 3-km lake areas in this local-area graphic is sharper is much improved over the previous version.

There are a few modifications to the text in this new version including a clarification on the meaning of the term 'cycling' to this: a continuous lake simulation forced by atmospheric conditions updated regularly by new atmospheric observations to obtain realistic lake water temperatures.

We hope this new version of our manuscript will be found as acceptable as a lake initialization paper in GMD.